# The Intravitreal Injection of Lanosterol Nanoparticles Rescues Lens Structure Collapse at an Early Stage in Shumiya Cataract Rats

**DOI:** 10.3390/ijms21031048

**Published:** 2020-02-05

**Authors:** Noriaki Nagai, Yuya Fukuoka, Kanta Sato, Hiroko Otake, Atsushi Taga, Mikako Oka, Noriko Hiramatsu, Naoki Yamamoto

**Affiliations:** 1Faculty of Pharmacy, Kindai University, 3-4-1 Kowakae, Higashi-Osaka, Osaka 577-8502, Japan; 1833420015f@kindai.ac.jp (Y.F.); 1645110002y@kindai.ac.jp (K.S.); hotake@phar.kindai.ac.jp (H.O.); punk@phar.kindai.ac.jp (A.T.); 2Faculty of pharmacy, Yokohama University of Pharmacy, Yokohama, Kanagawa 245-0066, Japan; m.oka@hamayaku.ac.jp; 3Laboratory of Molecularbiology and Histochemistry, Fujita Health University Institute of Joint Research, 1-98 Dengakugakubo, Kutsukake, Toyoake 470-1192, Aichi, Japan; norikoh@fujita-hu.ac.jp (N.H.); naokiy@fujita-hu.ac.jp (N.Y.)

**Keywords:** nanoparticle, lanosterol, cataract, intravitreal injection, lens

## Abstract

We designed an intravitreal injection formulation containing lanosterol nanoparticles (LAN-NPs) via the bead mill method and evaluated the therapeutic effect of LAN-NPs on lens structure collapse and opacification using two rat cataract models (SCR-N, rats with slight lens structure collapse; SCR-C, rats with the combination of a remarkable lens structure collapse and opacification). The particle size of lanosterol in the LAN-NPs was around 50–400 nm. A single injection of LAN-NPs (0.5%) supplied lanosterol into the lens for 48 h, and no irritation or muddiness was observed following repeated injections of LAN-NPs for 6 weeks (once every 2 days). Moreover, LAN-NPs repaired the slight collapse of the lens structure in SCR-N. Although the remarkable changes in the lens structure of SCR-C were not repaired by LAN-NP, the onset of opacification was delayed. In addition, the increase of cataract-related factors (Ca^2+^ contents, nitric oxide levels, lipid peroxidation and calpain activity levels) in the lenses of SCR-C was attenuated by the repeated injection of LAN-NPs. It is possible that a deficiency of lanosterol promotes the production of oxidative stress. In conclusion, it is difficult to improve serious structural collapse with posterior movement of the lens nucleus with a supplement of lanosterol via LAN-NPs. However, the intravitreal injection of LAN-NPs was found to repair the space and structural collapse in the early stages in the lenses.

## 1. Introduction

Vision impairment is a serious public health concern, and cataracts are a leading cause of blindness worldwide. Cataracts are characterized by opacification or optical dysfunction of the lens. With age, the prevalence of cataracts increases dramatically from 47% in those aged 55–64 years to 88% in those older than 75 years [1,2]. People suffering from cataracts undergo surgical treatment, and cataract surgeries account for a significant portion of healthcare costs due to the sheer prevalence of the disease among ageing populations in developed nations. On the other hand, there is major morbidity associated with cataracts in developing countries, where there is limited access to surgical care. Therefore, the development of effective and safe anti-cataract drugs is urgently required; however, a potent anti-cataract drug for human cataracts has not yet been introduced in a clinical setting.

In the lipid biosynthetic pathway, lanosterol is the first sterol and is initially converted by acetyl-CoA. Lanosterol synthase, 3-hydroxy-3-methylglutaryl-coenzyme A reductase, and squalene epoxidase are involved the complex process of lanosterol synthesis. Lanosterol has been discovered in human genetics studies to solubilize crystallin aggregates in vitro and in cells and is also able to restore lens transparency in rabbits and dogs with age-related cataracts [3]. However, a recent report showed that culturing 25 mM lanosterol (approximately 1%) with lens nuclei from 40 age-related cataractous human lenses for 6 days at room temperature failed to dissolve the aggregated proteins or restore clarity to the lens nuclei [4,5]. Thus, it is unclear if lanosterol ameliorates lens clarity by increasing the ability to physically dissolve protein aggregates and/or the denatured amyloid-like fibril proteins present in cataractous lenses. On the other hand, Kang et al. reported that lanosterol disrupts the aggregation of human γD-crystallin by binding to the hydrophobic dimerization interface and is likely correlated with oxidative stress processes [6]. In addition, lanosterol can play a protective role during the early stages of crystallin denaturation and lens epithelial cell apoptosis through oxidative-damage-induced UV-B exposure [7]. Huang et al. also reported that a combination of lanosterol and hesperetin is effective in delaying selenite-induced cataracts [8]. Therefore, the development of ophthalmic formulations containing lanosterol is expected in the ophthalmic field.

It is difficult to design ophthalmic formulations containing lanosterol that deliver lanosterol into the lens since lanosterol has limited solubility. Several advanced drug delivery systems (DDSs) have been designed and tested, such as liposomes, solid nanoparticles (NPs), polymeric NPs, nanomicelles, and nanospheres [9,10,11]. Recently, the biodegradable polymer most commonly used in drug-release systems was poly(lactic-co-glycolic acid) (PLGA); PLGA is also the preferred polymer in the synthesis of nano DDSs. Chitosan-coated liposomal formulations encapsulating a combination of lanosterol and hesperetin were also prepared by Huang et al. [8]. Thus, DDSs using nanotechnology are useful for designing formulations of insoluble drugs. In a previous study, we also designed solid nanoparticles created by a break-down mill (bead mill method) and reported the properties of low cell toxicity and high drug tissue uptake for the formulations containing solid nanoparticles compared to traditional formulations (liquid formulations and suspensions) [12,13,14,15,16]. In the future, solid nanoparticles may provide a novel strategy for developing safe and effective ophthalmic formulations.

The Shumiya cataract rat (SCR) is a hereditary cataractous rat strain in which the appearance of cataracts showing excessive collapse of the lens structure occurs in 66.7% of adults (cataractous SCR, SCR-C, ctr1 × ctr1, Ctr2 × Ctr2^l^) [17]. On the other hand, the remaining rats (33.3%, non-cataract SCR, SCR-N, ctr1 × ctr1, Ctr2 × Ctr2) have transparent lenses, although their lens structure still shows a slight collapse with aging compared to normal rats, such as Wistar rats and SD rats. In the present study, we attempt to develop an intravitreal injection formulation containing lanosterol nanoparticles (LAN-NPs) and evaluate the therapeutic effect of LAN-NPs on lens structure collapse and opacification in SCR-N and SCR-C.

## 2. Results

### 2.1. Design of Dispersions Containing Lanosterol Nanoparticles

Our previous study showed that it was possible to prepare insoluble-drug nanoparticle dispersions via a bead mill with methylcellulose type SM-4 (MC) and 2-hydroxypropyl-β-cyclodextrin (HPβCD) [12,13,14,15,16]. In this study, we undertook the design of dispersions containing lanosterol nanoparticles according to the previous method; however, the bead mill process resulted in the lanosterol reaching a meringue state (a mixture of air, water, and lanosterol). Therefore, we attempted to improve the bead mill method. In the observation of the meringue state, we found that the meringue state containing lanosterol returned to a dispersion when it was shaken lightly after the addition of saline (2000 rpm, 30 s, 4 °C). From these findings, we added this process to the bead mill protocol. The size of the lanosterol particles decreased by this new treatment to a range of 50–400 nm, with the mean particle size determined to be 191 and 125.2 nm by a laser diffraction particle size analyzer SALD-7100 and dynamic light scattering NANOSIGHT LM10, respectively (Figure 1). The solubility of lanosterol in these dispersions was 25.0 ± 0.19 µM, and most of the lanosterol in the dispersions was of a solid type (Figure 1D). Otherwise, the aggregation of lanosterol (LAN) particles was not detected for 2 weeks in the 0.5% LAN-NPs, although the LAN particles were aggregated 1 month after preparation in the LAN-NPs, and the zeta potential was 3.87 mV.

### 2.2. Safety Evaluation of Intravitreal Injections of LAN-NPs

Figure 2A shows the toxicity of the LAN-NPs on the human lens epithelial cell line SRA 01/04 (HLE cell). No serious cell damage was observed in the HLE cells exposed to 0.5% LAN-NPs. Next, we examined eye toxicity after the intravitreal injection of LAN-NPs. There were no apparent abnormalities seen in the eyes upon visual review (Figure 2B). In addition, Scheimpflug slit images obtained after the intravitreal injection of LAN-NPs showed no opacity or muddiness (Figure 2C). Next, we investigated the lanosterol concentration in rat lenses after the intravitreal injection of LAN-NPs (Figure 2D). Enhanced lanosterol levels in the lenses were observed and persisted over 48 h (lanosterol content in the non-instilled SCR-N, 1.78 ± 0.13 nmol, *n* = 5). The lanosterol level in the lenses of rats injected with LAN-NPs intravitreally was approximately 10 nmol/lens at 3 h after injection and subsequently decreased (Figure 2D). Changes in plasma lanosterol levels were not detected by a simple liquid chromatography with charged aerosol detector (LC-CAD) method. In addition, the lanosterol content in the lenses of rats injected with LAN-NPs intravitreally was higher than that in the instillation and intracameral injection (lanosterol content at 12 h after administration; instillation 1.79 ± 0.15 nmol/lens, intracameral injection 1.99 ± 0.20 nmol/lens, *n* = 3).

### 2.3. Changes in Lanosterol Levels, Opacity, and Structure in the Lenses of SCR-N and SCR-C with Aging

Figure 3 shows the lanosterol contents in, and the opacity and hematoxylin and eosin (H&E) images of, the lenses of SCR-N and SCR-C. Cataractous SCR (SCR-C) is the model animal used to study cataracts. The lanosterol levels in both SCR-N and SCR-C decreased with age, with the lanosterol levels in SCR-C significantly lower than those in SCR-N (Figure 3A). Moreover, lens opacification in SCR-C also occurred (Figure 3B,C), with posterior movement of the lens nucleus in SCR-C first observed at 12 weeks of age (Figure 3D). In contrast to the results in SCR-C, SCR-N showed no opacification at 12 weeks of age (Figure 3B,C), and the changes in the lens structure were minimal in comparison with SCR-C (Figure 3D).

### 2.4. Therapeutic Potential of Lanosterol in SCR-N and SCR-C Injected Intravitreally with LAN-NPs

Figure 4 shows the preventive effect of LAN-NPs on lens structure collapse in SCR-N. In the equator lentis, a slight space and structure collapse was observed at 8 and 10 weeks of age, and this structural change was impaired by the intravitreal injection of LAN-NPs. In this study, we also examined whether LAN-NPs can reverse lens opacification using SCR-C (Figure 5). Following the repetitive injection of LAN-NPs for 6 weeks, no inflammation was observed in the eyes of SCR-C (Figure 5A), and the progress of lens opacification was retarded (Figure 5B,C). However, the progression of lens opacification was not stopped, and lens opacification was observed in 12-week-old SCR-C intravitreally injected with LAN-NPs. In addition, remarkable changes in the lens structure with posterior movement of the lens nucleus were still observed in the intravitreally-injected SCR-C (Figure 5D). Moreover, the spatial and structural collapse and opacification in the lenses of SCR-C were also not repaired by the long-term treatment of LAN-NPs; the spatial and structural collapse and opacification were similar to the results found in 12- and 25-week-old SCR-C intravitreously injected with LAN-NPs (Figure 5E). It is known that Ca^2+^ content, Ca^2+^-ATPase activity, NO level, and LPO levels are related to lens opacification in SCR-C [18]. Therefore, we evaluated the changes in these factors in 11-week-old SCR-C to observe any preventive effects against lens opacification by LAN-NPs (Figure 6). Ca^2+^ contents, NO levels, and LPO and calpain activity levels were significantly enhanced, and Ca^2+^-ATPase activity decreased, in 11-week-old SCR-C, and the intravitreal injection of LAN-NPs attenuated these changes.

## 3. Discussion

Recently, it was reported that the administration of lanosterol plays a protective role in the early stages of crystallin denaturation and lens epithelial cell apoptosis by oxidative-damage-induced UV-B exposure [7]. Huang et al. showed that a combination of lanosterol and hesperetin is effective in delaying selenite-induced cataracts [8]. However, lanosterol has limited solubility, and it is difficult to design ophthalmic formulations that can deliver lanosterol into the lens. In order to overcome these difficulties, we investigated whether nanodispersions prepared by a bead mill can be used as an ophthalmic formulation for cataract treatment. We designed an intravitreal injection formulation containing lanosterol nanoparticles (LAN-NPs) and showed that these LAN-NPs impair the spatial and structural collapse that occurs in the early stages of cataract formation, thus delaying the onset of opacification in SCR.

First, we attempted to develop LAN-NPs. Additives are important for the preparation of solid nanoparticles by the bead mill method. Previous studies showed that adsorption to the surface of 5% cyclodextrin decreases the cohesion of nanoparticulate solids [19], and 0.5% MC is necessary for the preparation of drug solid nanoparticles by the bead mill method [20]. Based on these previous findings, we selected both MC and cyclodextrin as additives. However, lanosterol becomes meringue-like when subjected to 10 repeats of the bead mill method in the presence of 0.5% MC and 5% HPβCD (5500 rpm, 30 s, 4 °C). In our previous study, a meringue state was often observed under the bead mill method for cases of highly insoluble drugs, such as indomethacin. This state could be improved by increasing the MC content [20]. However, in the case of lanosterol, the meringue state of the dispersions was not resolved by increasing the MC content (0–5%), so other innovations to regulate the lanosterol particles were required. In this study, we found that gentle shaking after the addition of saline dispelled the meringue state, so this process was added to the bead mill treatment (the improved protocol). The lanosterol powder was changed to nanoparticles through the improved protocol, with particle sizes in the range of approximately 50–400 nm (Figure 1). Otherwise, we evaluated the dispersion stability. Although the aggregation of lanosterol particles was not detected for 2 weeks in the 0.5% LAN-NPs, the LAN particles were aggregated 1 month after their preparation in the 0.5% LAN-NPs. The zeta potential value is a major factor related to dispersion stability, but the value was not high enough (−3.87 mV). In a further study, the enhancement of zeta potential may be useful to improve the dispersion stability of LAN-NPs.

Next, we investigated whether LAN-NPs could be applied as an intravitreal injection. For the additives used, MC is highly bio-compatible, with low toxicity [21,22,23], and HPβCD is also known to cause no eye irritation at levels less than 12.5% in the eye membrane [24]. In this study, no damage was observed to lens epithelial cells exposed to LAN-NPs (Figure 2A), and no irritation, opacity, or muddiness were observed after the intravitreal injection of LAN-NPs in rats (Figure 2B,C). In addition, increased lanosterol levels were detected for more than 48 h after the injection of LAN-NPs (Figure 2D), and it is possible to maintain high lanosterol content (compared to a normal lens) in the lenses of the rats via the intravitreal injection of LAN-NPs (once every 2 days). Furthermore, we attempted to compare the safety between LAN-NPs and the solution containing lanosterol. The solution containing 0.5% lanosterol as dissolved using 15% DMSO, 20% ethanol, and/or a surface-active agent, such as tween 80, which was injected. However, the intravitreal injection of 0.5% lanosterol solution caused muddiness and promoted the risk of lens opacification. Further, it is difficult to apply lanosterol as an intravitreal injection formulation. In addition, we tried an injection of dispersions containing lanosterol microparticles, but the lanosterol suspensions also caused transient opacity after intravitreal injection, since their particle sizes were too large for use as an intravitreal injection. These results show that LAN-NPs are safe and that formulations meant to be injected intravitreally can be used. 

In studies to develop agents to protect against the collapse of the lens structure and lens opacification, the selection of experimental animals is very important. SCR-N and SCR-C are a hereditary cataractous rat strain, obtained by cross-breeding a spontaneous hypertensive rat and a Zucker fatty rat [25]. Although the lenses of SCR-N (which lack the mutations causing premature cataracts) remain transparent past 11 weeks of age, the lens structures at 8–12 weeks of age are unstable compared to normal rats, such as Wistar rats and SD rats [25]. On the other hand, SCR-C carries a specific combination of hypomorphic mutations of lanosterol synthase and FDFT1 (farnesyl-diphosphate-farnesyl-transferase-1) genes, and the mutation of the lanosterol synthase gene results in decreased cholesterol levels in the lens, causing cataracts [25]. Opacity from the perinuclear zone to the cortical intermediate layer is observable at around 11 weeks of age [17,26,27,28]. In addition, the extension of anterior sutural hypoplasia is observed at the onset of cataracts, and it is known that the morphological mechanism in SCR-C causes liquefied anterior cortical fibers toward the posterior subcapsular region [26,28]. In this study, the lanosterol levels in the lenses of SCR-N tended to decrease with age, and the lens structures showed a slight collapse at 8–12 weeks of age, although lens opacification was not observed (Figure 3 and Figure 4). In contrast to the results for SCR-N, the lanosterol levels in the lenses of SCR-C were obviously decreased, and posterior movement of the lens nucleus and lens opacification were observed at 12 weeks of age (Figure 3). Thus, we concluded that SCR-N and SCR-C present a plausible model system for studying the effectiveness of LAN-NPs for the treatment of cataracts, thereby demonstrating the therapeutic potential of intravitreal injections of LAN-NPs in SCR-N and SCR-C.

In SCR-N subjected to repetitive intravitreal injections of 0.5% LAN-NPs, the slight spatial and structural collapse seen at 8 weeks of age was impaired (Figure 4). Moreover, the repetitive intravitreal injection of LAN-NPs delayed the onset of lens opacification in the SCR-C (Figure 5B,C). Zhao et al. investigated lanosterol itself as a potential treatment for cataracts and showed that the intravitreal injection of 2 mg/mL lanosterol every 3 days can reverse crystallin aggregation in dogs [3]. It was also reported that lanosterol disrupts the aggregation of human γD-crystallin by binding to the hydrophobic dimerization interface [6]. Furthermore, lanosterol has been shown to play a protective role in the early stages of crystallin denaturation and lens epithelial cell apoptosis by oxidative-damage-induced UV-B exposure [7]. Additionally, the combination of lanosterol and hesperetin is effective in delaying the onset of selenite-induced cataracts [8]. On the other hand, a previous study also reported that lanosterol increases lens transparency in both rabbit cataractous lenses and in vivo dog experiments [3]. These slight structural changes were repaired in our study using SCR-N (Figure 4). However, the serious structural collapses with posterior movement of the lens nucleus in the SCR-C were not reversed, and no increase in the transparency of opaque lenses were observed in SCR-C (Figure 5B–E). This discrepancy may be caused by the differences in the models, since the structural changes in the lenses of the dogs and rabbits used by Zhao et al. were mild in comparison with those of the SCR-C [3]. In contrast to the results of Zhao et al. [3], Daszynski et al. failed to find evidence that lanosterol has either anti-cataractogenic activity or binds aggregated lens proteins to dissolve cataracts [4]. Taken together, it was suggested that the difference in the severity and part (cortex and nucleus) of the lens opacification may explain the contradiction in the previous reports for therapeutic effect by lanosterol [3,4,5], and our study may reflect both of these studies’ results. Further studies are needed to elucidate the relationships between the effects of repair and structural collapse by lanosterol and to evaluate whether the application of lanosterol can return opaque lenses to transparency.

There have been many studies concerning the mechanism of lens opacification in SCR-C, with results that can be summarized as follows [22,29,30]. Excessive NO, a product of oxidative stress, causes enhanced lipid peroxidation that results in the oxidative inhibition of Ca^2+^-ATPase. The decrease in Ca^2+^-ATPase activity induces the elevation of lens Ca^2+^ levels and activates calpain. Thereafter, the degradation of lens proteins, such as crystallin proteins, by calpain results in lens opacification [18]. Ca^2+^ content, NO levels, LPO, and calpain activity levels are enhanced, and Ca^2+^-ATPase activity is decreased in SCR-C (Figure 6); these changes are attenuated by the intravitreal injection of LAN-NPs (Figure 6). From these results, we demonstrated the effect of lanosterol on NO release using an HLE cell model stimulated by interferon-γ (IFN-γ) and lipopolysaccharide (LPS) [31,32]. Lanosterol treatment had no effect on the changes in NO release caused by the stimulation of HLE cells by IFN-γ and LPS (non-treated HLE cells, 0.09 ± 0.01 nmol/10^6^ cells; HLE cells stimulated with IFN-γ and LPS, 2.10 ± 0.29 nmol/10^6^ cells; LAN-NP-treated HLE cells stimulated with IFN-γ and LPS, 2.05 ± 0.27 nmol/10^6^ cells; *n* = 7–12). Taken together, we hypothesize that the spatial and structural collapse in the lens causes NO production and that the accelerated production of oxidative stress may enhance lens opacification via calpain activation.

Further studies are needed to establish a preparation method for an intravitreal injection formulation containing higher lanosterol concentrations. Furthermore, it is very important to design an eyedrop formulation that can deliver lanosterol into the lens. We previously reported that solid drug nanoparticles in ophthalmic formulations are taken up into the corneal epithelium by energy-dependent endocytosis and show high rates of transcorneal penetration [12,33,34]. Therefore, we are now planning to design eyedrops containing solid lanosterol nanoparticles and investigate their therapeutic effects by injecting solid lanosterol nanoparticles in SCR-N and SCR-C.

## 4. Materials and Methods 

### 4.1. Animals

Male SCR-N (non-cataract SCR) and SCR-C (cataractous SCR) were housed in accordance with the Pharmacy Committee Guidelines for the Care and Use of Laboratory Animals at the Yokohama University of Pharmacy. The experiments were performed at Kindai University and approved on 1 April 2013 (project identification code, KAPS-25-003) according to the Pharmacy Committee Guidelines for the Care and Use of Laboratory Animals in Kindai University. The concentrations and numbers of doses of lanosterol were based on those of the previous study by Zhao et al. (using dogs) [3]; 0.5% lanosterol (5 µL) was injected into the vitreous of the right eye of 6-week-old SCR for 6 weeks (once every 2 days) under isoflurane anesthesia. All experiments were performed in accordance with the guidelines for the Association for Research in Vision and Ophthalmology (ARVO).

### 4.2. Chemicals

The 2-hydroxypropyl-β-cyclodextrin (HPβCD) was obtained from Nihon Shokuhin Kako Co., Ltd. (Tokyo, Japan). Isoflurane, mannitol, and Ca Test Kits were purchased from Wako Pure Chemical Industries, Ltd. (Osaka, Japan). Methylcellulose type SM-4 (MC) was supplied by Shin-Etsu Chemical Co., Ltd. (Tokyo, Japan). Lanosterol and the LPO Assay Kit (BIOXYTECH^®^ LPO-586^TM^) were purchased from Nacalai Tesque, Inc. (Kyoto, Japan) and OXIS International, Inc. (Portland, OR, USA), respectively. The Bio-Rad Protein Assay Kit was provided by Bio-Rad Laboratories (Hercules, CA, USA). Benoxil (0.4%) and pivalephrine (0.1%) were obtained from Santen Pharmaceutical Co., Ltd. (Osaka, Japan) and Kanto Chemical Co., Inc. (Tokyo, Japan), respectively. Calpain Activity Fluorometric Assay Kits were provided by BioVision Inc. (San Francisco, CA, USA). All other chemicals were of the highest purity commercially available.

### 4.3. Preparation of LAN-NPs

LAN-NPs (0.5%, pH 7.0) were prepared according to our previous reports using a Bead Smash 12 (Wakenyaku Co. Ltd., Kyoto, Japan) [12,13,14,15,16]. The lanosterol powder (0.5%) was dispersed in saline with 0.5% MC and 5% HPβCD and milled with 0.1 mm zirconia beads as follows: (i) 5500 rpm, 30 s × 20 times, 4 °C; (ii) addition of saline with MC and HPβCD; (iii) 2000 rpm, 30 s × 3 times, 4 °C; (iv) 5500 rpm, 30 s × 10 times, 4 °C; (v) 2000 rpm, 30 s × 7 times, 4 °C.

### 4.4. Measurement of Lanosterol by the LC-CAD Method

Lanosterol was measured by a simple liquid chromatography with charged aerosol detector) (LC-CAD) method consisting of a Shimadzu (Kyoto, Japan) Model LC-10AD pump, a Shimadzu Model DGU-12A degasser, and a Corona Veo detector (Thermo Fisher Scientific, Inc., Waltham, MA, USA). A TSK gel ODS-100S (5 µm, 150 × 2.0 mm I.D.) from Tosoh Co. (Tokyo, Japan) was chosen for the LC column. The flow rate was 0.4 mL/min, and 20 µL samples in methanol were injected. A step gradient experiment was performed using 5% and 100% methanol over 19 min. The measurement conditions of the CAD were as follows: highly pure nitrogen (99.9%) for the detector was produced by an AT-2NP-CAD (AIR-TEC Co, Kanagawa, Japan), with an inlet pressure (nitrogen) of 61.9 psi. The charger voltage was 2.66 kV, and the charger current was 0.99 µA.

### 4.5. Evaluation of the Characteristics of the LAN-NPs

Zeta potential was evaluated using a Zeta Potential Meter Model 502 (Nihon Rufuto Co., Ltd., Tokyo, Japan). The particle size distribution was measured by a laser diffraction particle size analyzer SALD-7100 (Shimadzu Corp., Kyoto, Japan; refractive index 1.60-0.10i) and dynamic light scattering NANOSIGHT LM10 (QuantumDesign Japan, Tokyo, Japan). In addition, atomic force microscope (AFM) images were obtained using a scanning probe microscope SPM-9700 (Shimadzu Corp., Kyoto, Japan). To measure solubility, LAN-NP samples were filtered through 25 nm pore size membrane filters (MF™-MEMBRANE FILRER, Merck Millipore, Tokyo, Japan), and the lanosterol concentration in the filtrate was determined by the LC-CAD method described above.

### 4.6. Measurement of Lanosterol content in Rat Lenses

Lanosterol content in the lens was measured as follows. The lens was homogenized in 400 µL methanol, and the homogenates were centrifuged at 20,400× *g* for 15 min at 4 °C. The supernatants were used for the measurement of lanosterol by the LC-CAD method described above.

### 4.7. Measurement of Toxicity using Cultured Human Lens Cells

The human lens epithelial cell line SRA 01/04 (HLE cell) was cultured under humidified air containing 5% CO_2_ at 37 °C, following our previous reports [31,35]. Briefly, HLE cells were cultured in Dulbecco’s modified Eagle’s medium with gentamicin (10 mg/L) and 10% (v/v) heat-inactivated fetal bovine serum. The HLE cells (1 × 10^4^ cells) in 96-well microplates (IWAKI, Chiba, Japan) were incubated for 3 days. After that, the cells were treated for 24 h by the LAN-NPs, and the cell viability was measured with a Cell Count Reagent SF, according to the manufacturer’s instructions. The cell viability is represented by the following equation (Equation (1)):Cell viability (%) = Abs_treatment_ / Abs_non-treatment_ × 100 (1)

### 4.8. Scheimpflug Slit Images in the SCR

The SCR without anesthesia were dilated by 0.1% pivalephrine and monitored by EAS-1000 (Nidek, Aichi, Japan). The changes in the transparency of their lenses was analyzed following our previous report [36]. The total area of the opacity of the lenses was expressed as pixels (thread level 100, flash level 100, slit length 4.2 mm).

### 4.9. Hematoxylin and Eosin (H&E) Staining of the Lens

SCR were killed by injecting a lethal dose of pentobarbital. Then, their eyes were removed and fixed in a SUPER FIX™ rapid fixative solution. Next, 3 µm paraffin sections were prepared and stained with H&E [37].

### 4.10. Measurement of Cararact-Related Factors

The SCR were killed by injecting a lethal dose of pentobarbital. Their eyes were removed and stored at −80 °C until being used for the measurement of Ca^2+^ content, Ca^2+^-ATPase activity, nitric oxide (NO) level, lipid peroxidation (LPO), and calpain activity level. The Ca^2+^ content in the lenses was determined using a Ca Test Kit (methyl xylenol blue colorimetric method) [18], and Ca^2+^-ATPase activity was calculated as the difference in the Pi liberated from the ATP measured in the presence and absence of Ca^2+^ [36]. NO levels were determined using a concentric microdialysis probe (A-1-20-05, 5 mm length; Eicom, Kyoto, Japan) and a flow-through spectrophotometer (NOD-10, Eicom, Kyoto, Japan) [31]. In this study, the amounts of NO reflect the level of the NO_2_^−^ metabolite, which is produced from NO. LPO levels were measured by measuring the lipid peroxidation products malondialdehyde and 4-hydroxynonenal using an LPO Assay Kit according to the manufacturer’s instructions [18]. The protein levels in the samples used to determine Ca^2+^-ATPase activity and LPO levels were assessed using a Bio-Rad Protein Assay Kit. In addition, the calpain activity was measured at 505 nm by a Calpain Activity Fluorometric Assay Kit according to the manufacturer’s instructions. The calpain activity was represented as the ratio of calpain activity levels in 6-week-old SCR-C (relative calpain activity).

### 4.11. Statistical Analysis

Data are expressed as the mean ± standard error (S.E.) of the mean. Statistical significance (*p* < 0.05) was evaluated using Student’s *t*-test and ANOVA followed by Dunnett’s multiple comparison.

## 5. Conclusions

We designed solid lanosterol nanoparticles and found that they can be administered as intravitreal injections. It is difficult to improve serious structural collapse with posterior movement of the lens nucleus and reverse opacification by the supplement of lanosterol. However, supply of lanosterol by the intravitreal injection of solid lanosterol nanoparticles repaired the spatial and structural collapse in the lenses of SCR-N during the early stages. These findings suggested that the difference in the severity and part (cortex and nucleus) of the lens opacification may explain the contradiction in the previous reports of lanosterol as a cataract inhibitor or preventing compound [3,4,5]. In addition, we found that the spatial and structural collapse in the lens was related the production of oxidative stress, e.g., NO production. These findings contribute to the design of anti-cataract drugs.

## Figures and Tables

**Figure 1 ijms-21-01048-f001:**
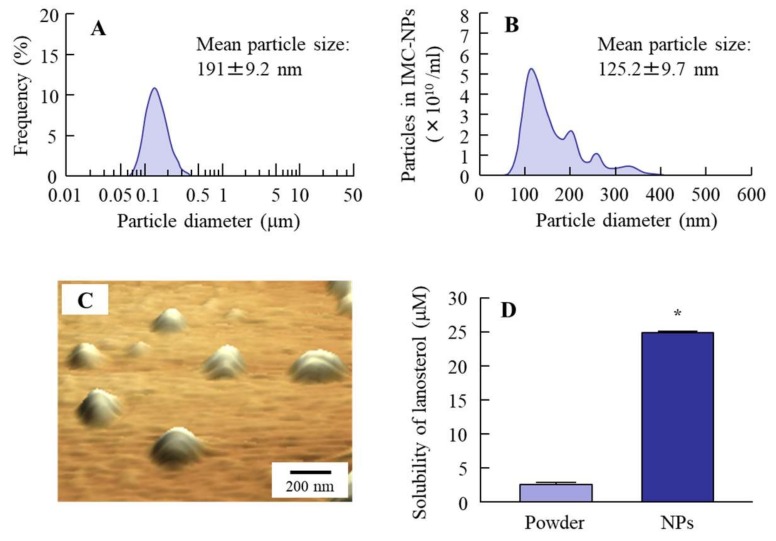
Particle size, atomic force microscope (AFM) image, and solubility of lanosterol nanoparticles prepared by bead mill treatment. (**A**,**B**) The particle size frequencies of lanosterol nanoparticles (LAN-NPs) obtained by the SALD-7100 (A) and NANOSIGHT LM10 (B). (**C**) An AFM image of LAN-NPs. (**D**) The solubility of lanosterol in the LAN-NPs. *n* = 5. * *p* < 0.05, vs. lanosterol (LAN) powder. The particle size of lanosterol was decreased to a nano size by the bead mill method, and the ratio of solid- to solution-type lanosterol was 99.8:0.2 in the 0.5% LAN-NP formulation.

**Figure 2 ijms-21-01048-f002:**
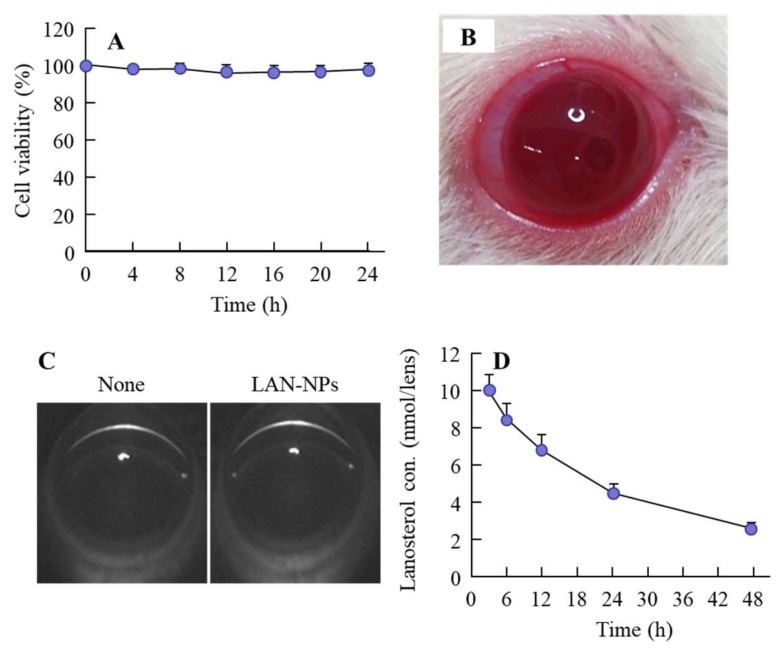
Potential adaptability of LAN-NPs as an intravitreal injection formulation. (**A**) Viability of HLE cells treated with LAN-NPs. (**B**,**C**) Eye image (B) and Scheimpflug slit images (C) of rats immediately after the intravitreal injection of LAN-NPs. (**D**) Changes in lanosterol concentration in the lenses of rats intravitreously injected with LAN-NPs. None, non-injected rats with slight lens structure collapse (SCR-N); LAN-NPs, LAN-NP injected SCR-N; *n* = 6. No cell toxicity was observed after treating human lens epithelial cell line SRA 01/04 (HLE) with LAN-NPs, and no inflammation or opacity was observed after intravitreal injection in SCR-N. Lanosterol levels in the lenses retained their increase for 0–48 h following the intravitreal injection of LAN-NPs.

**Figure 3 ijms-21-01048-f003:**
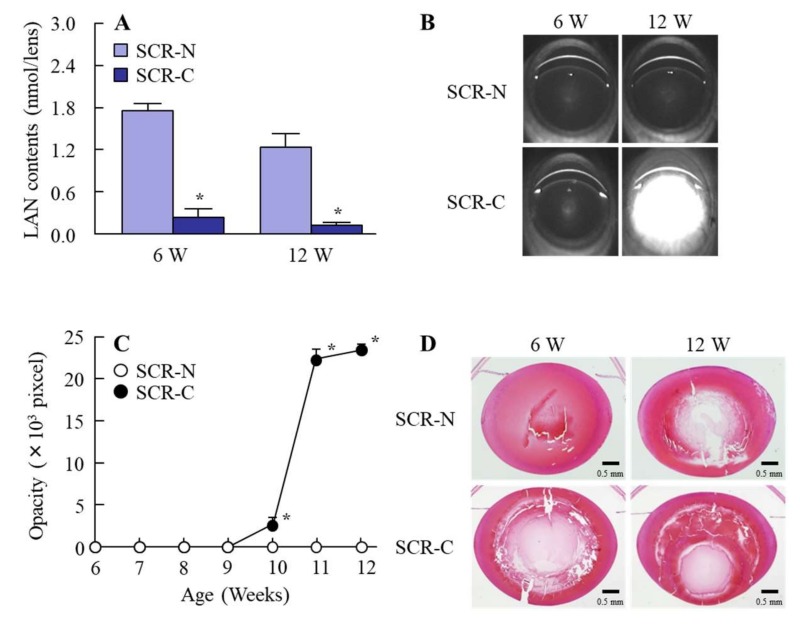
Lanosterol contents, opacity levels, and H&E images of the lenses of SCR-N and with the combination of a remarkable lens structure collapse and opacification (SCR-C) aged 6–12 weeks. (**A**) Lanosterol contents in the lenses of SCR-N and SCR-C. (**B**,**C**) Scheimpflug slit images (B) and opacification levels (C) in SCR-N and SCR-C. (**D**) H&E stained lens images of SCR-N and SCR-C. *n* = 5–8. * *p* < 0.05 vs. SCR-N for each group. The lanosterol levels in SCR-C were lower than those in SCR-N, and the lens opacification and posterior movement of the lens nuclei were observed in 12-week-old SCR-C.

**Figure 4 ijms-21-01048-f004:**
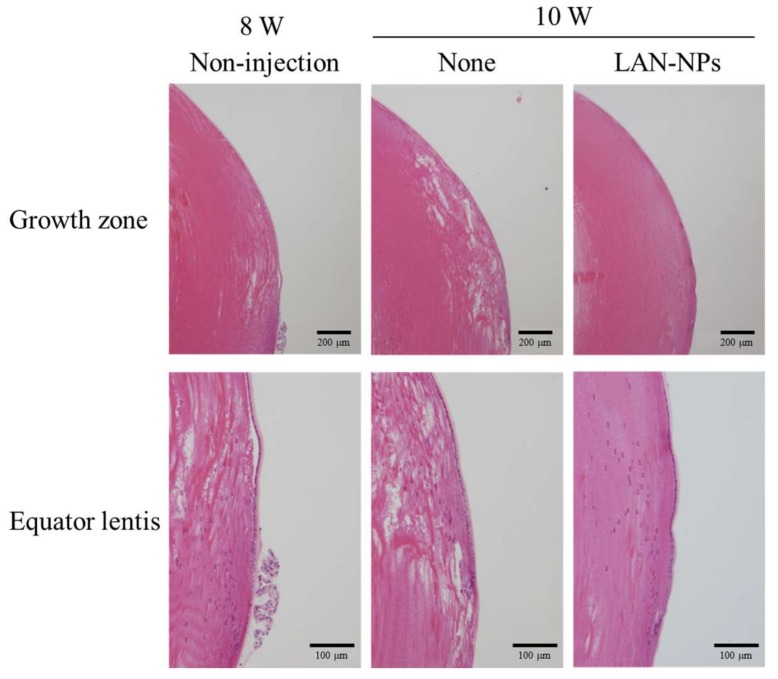
Changes in the lens structure of the growth zone and equator lentis in 8- and 10-week-old SCR-N intravitreously injected or not injected with LAN-NPs. Eight week-old SCR-N were intravitreously injected with LAN-NPs for 2 weeks (once per 2 days). Non-injection, non-injected SCR-N aged 8 weeks; None, non-injected SCR-N aged 10 weeks; LAN-NPs, LAN-NP injected SCR-N aged 10 weeks. Slight collapse of the lens structure was observed in the SCR-N aged 8 and 10 weeks. On the other hand, the intravitreal injection of LAN-NPs reversed spatial and structural collapse in the lenses of SCR-N.

**Figure 5 ijms-21-01048-f005:**
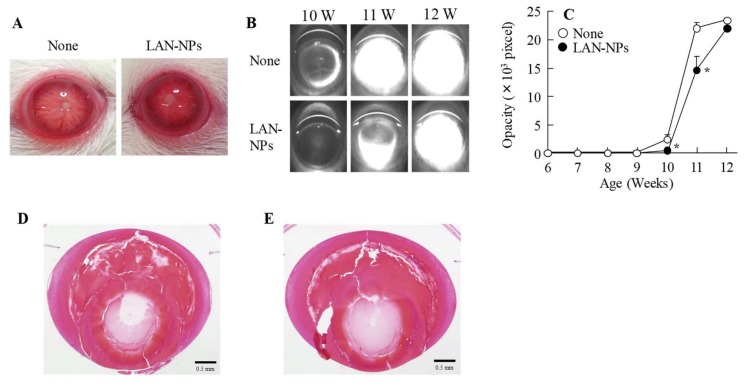
Eye images, Scheimpflug slit images, opacity levels, and H&E image of SCR-C after repetitive injection of LAN-NPs for 6 weeks. LAN-NPs were intravitreously injected into 6-week-old SCR-C, and the injections were continued for 6 weeks (once per 2 days). (**A**) Eyes of SCR-C intravitreously injected with or without LAN-NPs. (**B**,**C**) Scheimpflug slit (B) and opacification levels in SCR-C intravitreously injected with or without LAN-NPs. (**D**) H&E image of 12-week-old SCR-C intravitreously injected with LAN-NPs. (**E**) H&E image of 25-week-old SCR-C intravitreously injected with LAN-NPs. None, non-injected SCR-C; LAN-NPs, LAN-NP injected SCR-C; *n* = 6. * *p* < 0.05, vs. None for each group. No eye inflammation was observed in the SCR-C after the repetitive injection of LAN-NPs. LAN-NPs delayed but did not prevent the onset of opacification and showed remarkable changes in the lens structure, such as the posterior movement of the lens nucleus in SCR-C intravitreously injected with LAN-NP.

**Figure 6 ijms-21-01048-f006:**
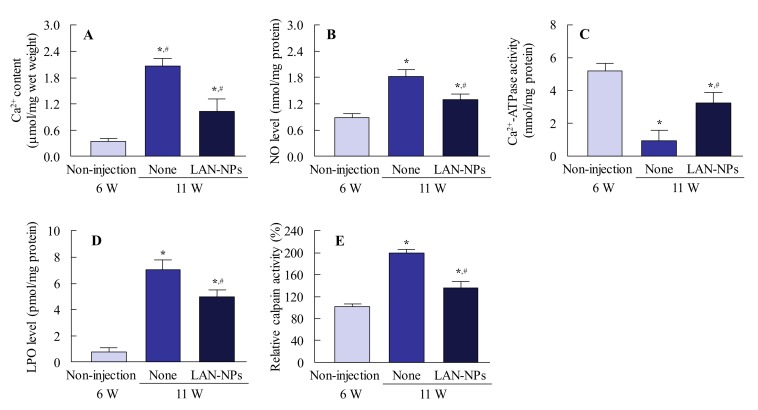
Changes in Ca^2+^ contents, NO levels, Ca^2+^-ATPase activity, and LPO levels in the lenses of 11-week-old SCR-C intravitreously injected with LAN-NPs. LAN-NPs were injected intravitreously into 6-week-old SCR-C for 5 weeks (once per 2 days). (**A**) Ca^2+^ contents in the lenses of SCR-C with or without LAN-NP injection. (**B**) NO levels in the lenses of SCR-C with or without LAN-NP injection. (**C**) Ca^2+^-ATPase activity in the lenses of SCR-C with or without LAN-NP injection. (**D**) LPO levels in the lenses of SCR-C with or without LAN-NP injection. (**E**) Calpain activities in the lenses of SCR-C with or without LAN-NPs injection. Non-injection, non-injected SCR-C aged 6 weeks; None, non-injected SCR-C aged 11 weeks; LAN-NPs, LAN-NPs injected SCR-C aged 11 weeks; *n* = 6–11; * *p* < 0.05, vs. Non-injection for each group. ^#^
*p* < 0.05, vs. None for each group. The changes in Ca^2+^ contents, NO levels, Ca^2+^-ATPase activity, LPO, and calpain levels in the lenses of SCR-C were attenuated by the repeated injection of LAN-NPs.

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
