# Peer review of "The Intravitreal Injection of Lanosterol Nanoparticles Rescues Lens Structure Collapse at an Early Stage in Shumiya Cataract Rats"

_ijms, 2020, doi:10.3390/ijms21031048_

Round 1

Reviewer 1 Report

the paper is interesting but is too specific for the reader materials sample of experiment and results may be presented in a more comprehensible form.

the study is really interesting but the reader need to evaluate potential developments of these data at least at speculative level.

Author Response

We carefully revised our manuscript according to the suggestions of the reviewer 1, and details are as follows.

< Q and A for Reviewer 1>

Q1. The paper is interesting but is too specific for the reader materials sample of experiment and results may be presented in a more comprehensible form. The reader need to evaluate potential developments of these data at least at speculative level.

A1. The reviewer’s comments are very important. We ordered a professional English editing service in the MDPI, and revised the manuscript to more comprehensible form. In addition, the purpose and novelty are more clearly expressed. Thank you very much for pointing this out.

Thank you for great comments.

Reviewer 2 Report

Shumiya rats develop cataract due to the disruption in lanosterol synthetic pathway. The authors show that supplementing the animals with lanosterol via injection into the vitreous, partially rescues the lens morphology and fiber cell arrangements. The lanosterol was administered in nanoparticle form, synthesized in a modified form. The lenses from the treated and control animals were analyzed by standard methods and the investigators found that supplementation of the animal eyes with lanosterol particles resulted in changes to Ca 2+ and NO levels and partiall restoration of the levels of LPO and Ca 2+ ATPase activity. However, the development of lens opacity was not affected significantly (Fig 3 and 5). In fact, at the end of study period of 12 weeks the opacity in treated and untreated lenses were similar. The slightly lower opacification observed on week 11 in treated eyes by the authors is therefore non-consequential. The partial restoration of lens morphology in this model is not surprising because the authors have simply provided the metabolite, lanosterol, that was deficient in the lenses of these animals due to a genetic defect. In spite of the negative effect on maintaining the lens transparency in lanosterol treated rats the authors suggest that lanosterol NP can be used to maintain lens transparency (in abstract and elsewhere). Clearly, the study results do not support this. Previous studies have also shown that supplementation of lanosterol to Shumiya rats alleviates some of the lens defects in these animals. Therefore, the study does not advance our understanding of Shumiya cataract or support the notion that lanosterol prevents lens opacity formation or cataractogenesis. There are also other concerns about the manuscript and those are listed below.

Abstract: lines 27 & 28-  see the comment above.

Introduction: lines 55-57 (ref 8). It appears the authors have misquoted/misunderstood the study and study results. In the cited study, Huang et al administered lanosterol + hesperitin as combination drugs to the rats treated with selenite and found that the combination therapy was effective in suppressing the cataract development. The selenite-induced cataract model is an oxidative stress-induced cataract model. Hesperitin is an anti-oxidative agent and it is not surprising that the Huang et al observed that the tested drug combination was effective. Huang’s study did not include lanosterol by itself or hesperitin by itself. Therefore, it is not correct to state “lanosterol was effective in delaying selenite-induced cataracts” in the study published by Huang et al.

Results: Line 94: The zeta value of -3.87 mV seems very low and the NP will be less stable in solution.

The authors should have measured calpain levels in the lenses (NP treated and untreated) and also analyzed the lens proteins to see whether the crystallin proteolysis by calpain was prevented in lanosterol-NP treated animals. These expts should be part of studies used to test the efficacy of drug candidates because in Shumiya rats the loss of membrane integrity due to impairment in sterols leads to leaky membranes, Ca2+ influx, calpain activation and proteolysis. An effective drug is expected to show prevention of calpain-induced proteolysis.

Discussion: line 243:  The results of this study do not support the view that lanosterol is effective in delaying cataract formation or helps to maintain lens transparency. To this reviewer it appears that this study contradicts the earlier claims that lanosterol can effectively delay cataract formation. This should be stated in the discussion and abstract section as well. The data shown in Fig 5 there is no difference in the opacity of treated and control animals at 12 weeks.  Long-term data is more important in the case of cataract prevention studies because cataract itself is a slow process.   

Author Response

We carefully revised our manuscript according to the suggestions of the reviewer 2, and details are as follows (we ordered a professional English editing service in the MDPI).

< Q and A for Reviewer 2>

Q1. Abstract: lines 27 & 28- The authors suggest that lanosterol NP can be used to maintain lens transparency (in abstract and elsewhere). Clearly, the study results do not support this.

A1. The reviewer’s comment is correct. We corrected the sentence to “it is difficult to improve serious structural collapse with posterior movement of the lens nucleus with a supplement of lanosterol via LAN-NPs. However, the intravitreal injection of LAN-NPs was found to repair the space and structural collapse in the early stages in the lenses” (line 30-35, 432-438).

Q2. The study does not advance our understanding of Shumiya cataract or support the notion that lanosterol prevents lens opacity formation or cataractogenesis.

A2. In this study, we designed solid lanosterol nanoparticles, and found they can be administered as intravitreal injections without transient opacity and inflammation. Moreover, we showed that it is difficult to improve the serious structural collapse with posterior movement of the lens nucleus, and reverse the opacification by the supplement of lanosterol, although, the supply of lanosterol by the intravitreal injection of solid lanosterol nanoparticles repaired the space and structural collapse in the early stages in the lenses of SCR-N. In addition, we found that the space and structure collapse in the lens accelerated to produce the oxidative stress, such as NO in the SCR-C. These findings have enough novelty, and contribute to the design of anti-cataract drugs, and support previous studies for anti-cataract effect of lanosterol.

Q3. Introduction: lines 55-57 (ref 8). It appears the authors have misquoted/misunderstood the study and study results. It is not correct to state “lanosterol was effective in delaying selenite-induced cataracts” in the study published by Huang et al.

A3. Thank you very much for pointing this out. In order to respond to the reviewer’s comment, we corrected the sentence to “Huang et al. also reported that a combination of lanosterol and hesperetin is effective in delaying selenite-induced cataracts” (line 65-66, 78, 216, 294).

Q4. Results: Line 94: The zeta value of -3.87 mV seems very low and the NP will be less stable in solution.

A4. The reviewer’s comments are very important. The zeta value is not enough levels, and need to enhance. In fact, the aggregation of LAN particles was not detected for 2 weeks in the 0.5% LAN-NPs, but the LAN particles was aggregated 1 month after the preparation in the 0.5% LAN-NPs. On the other hand, we prepared the solution containing 0.5% LAN by using the DMSO, ethanol and/or surface active agent, such as tween 80 in the preliminary study. The intravitreal injection of 0.5% LAN solution in DMSO, ethanol and/or surface active agent promote the onset of lens opacification in the SCR-C in comparison with non-treated SCR-C, and could not apply as intravitreal injection formulation. From these background, the LAN-NPs without serious side effect was expected to applicate the anti-cataract drugs than that in solution containing LAN. In order to respond to the reviewer’s comment, we added these contents in the discussion (line 107-110, 240-245, 254-262).

Q5. The authors should have measured calpain levels in the lenses (NP treated and untreated) and also analyzed the lens proteins to see whether the crystallin proteolysis by calpain was prevented in lanosterol-NP treated animals.

A5. Thank you very much for pointing this out. We measured the calpain activity in the lenses, and the calpain activity in the SCR-C applied with LAN-NPs was lower than that in non-treated SCR-C (Fig. 6E). Whereas, we don’t have the data for crystallin proteolysis, however previous reports showed that the crystallin proteolysis related to calpain activity (Ref 30,31), and we also showed the lens structure collapse of SCR-C in the H.E. image (Fig. 5). In order to respond to the reviewer’s comment, we added these additional data and contents (line 177, 206-207, 210, 313, 351-352, Figure 6E).

Q6. Discussion: line 243: The results of this study do not support the view that lanosterol is effective in delaying cataract formation or helps to maintain lens transparency. To this reviewer it appears that this study contradicts the earlier claims that lanosterol can effectively delay cataract formation. This should be stated in the discussion and abstract section as well. The data shown in Fig 5 there is no difference in the opacity of treated and control animals at 12 weeks. Long-term data is more important in the case of cataract prevention studies because cataract itself is a slow process.

A6. The reviewer’s comment is correct. In order to respond to the reviewer’s comment, we added the long-term data (Fig. 5E). The space and structural collapse and opacification in the lenses of SCR-C were not repaired by the long-term treatment of LAN-NPs, since the space and structural collapse and opacification were similar to 12 and 25 week-old SCR-C intravitreously injected with LAN-NPs. On the other hand, a slight space and structure collapse observed at 8 weeks of age were reversed by the intravitreal injection of LAN-NPs. We think that these data support previous studies. We added the additional result for long-term data, and revised the sentence (line 30-35, 171-174, 194-195, Figure 5E).

Thank you for great comments.

Reviewer 3 Report

The manuscript aims to report that intravitreal injection of lanosterol nanoparticles (LAN-NPs) rescues lens structure collapse at an early stage in shumiya cataract rats. In my opinion, numerous investigators have previously used lanosterol as a drug against cataract (please refer to the following papers: #1 Shanmugam PM, Barigali A, Kadaskar J, Borgohain S, Mishra DK, Ramanjulu R, Minija CK. Effect of lanosterol on human cataract nucleus. Indian J Ophthalmol 2015;63:888-890. #2 Zhao L, Chen XJ, Zhu J, Xi YB, Yang X, Hu LD, Ouyang H, Patel SH, Jin X, Lin D, Wu F, Flagg K, Cai H, Li G, Cao G, Lin Y, Chen D, Wen C, Chung C, Wang Y, Qiu A, Yeh E, Wang W, Hu X, Grob S, Abagyan R, Su Z, Tjondro HC, Zhao XJ, Luo H, Hou R, Jefferson J, Perry P, Gao W, Kozak I, Granet D, Li Y, Sun X, Wang J, Zhang L, Liu Y, Yan YB, Zhang K. Lanosterol reverses protein aggregation in cataracts. Nature 2015;523:607-611.). Although the authors hypothesized that solid nanoparticle form of lanosterol can increase the bioavailability of drugs that are not absorbed efficiently, their study design is not well organized to reflect the significance of nanoparticle formulations. Although the in vivo study demonstrates the use of LAN-NPs may be beneficial to cataract treatment, the lack of the most important control group (i.e., the same amount of LAN, but not in the form of NP) constitutes the major weakness here.

Another obvious drawback is that the authors should investigate the effect of concentration of LAN-NPs on the therapeutic efficacy of cataract. Otherwise, the audiences may be curious about why they intended to select specific dose of LAN-NPs for formulation design. At the current stage, this work does not reach the required level for publication in a high-quality journal “IJMS” and seems difficult to be improved through standard revisions.

Specific comments:

The intravitreal injection is a commonly used technique to deliver medications to the posterior segment of the eye. In my opinion, the in vivo fate of injectables is particularly important for clinical end user. The authors simply examine the therapeutic efficacy of the injected LAN-NPs, but do not check the biodistribution of the drugs. It is also necessary to investigate the relationship between in vivo amount of LAN-NPs and residence time. If the authors consider the nanoparticle formulation can improve ocular bioavailability, they should clarify the underlying mechanism. It is also highly desirable to investigate whether the physicochemical properties of nanoparticles play any potential roles in the control of their interactions with biological entities in vivo? The authors should check these parts by comprehensively performing cell-material interaction studies using a series of LAN-NPs with varying sizes and charges. As stated by the authors, there have been many studies concerning the mechanism of lens opacification in SCR, with results that can be summarized as follows: excessive NO, a product of oxidative stress, causes enhanced lipid peroxidation that results in the oxidative inhibition of Ca2+-ATPase. Please add appropriate citations to support this important claim. In the results section, the authors mentioned the zeta potential value and particle size of nanoparticles, but in the discussion section, they did not describe the influence of these parameters on disease treatment. As stated by the authors, the supply of lanosterol by the intravitreal injection of LAN-NPs repaired the space and structural collapse in the early stages in the lenses of SCR-N and delayed the progress of lens opacification in SCR-C. In order to publish in IJMS, it is highly desirable to investigate the above-mentioned molecular mechanism by designing corresponding experiments.

Author Response

We carefully revised our manuscript according to the suggestions of the reviewer 3, and details are as follows (We ordered a professional English editing service in the MDPI).

<Q and A for Reviewer 3>

Q1. Although the in vivo study demonstrates the use of LAN-NPs may be beneficial to cataract treatment, the lack of the most important control group (i.e., the same amount of LAN, but not in the form of NP) constitutes the major weakness here.

A1. The reviewer’s comments are very important. In the preliminary study, we attempted to prepare the solution containing 0.5% LAN by using the DMSO, ethanol and/or surface active agent, such as tween 80, however, the intravitreal injection of 0.5% LAN solution in DMSO, ethanol and/or surface active agent promote the onset of lens opacification in the SCR-C in comparison with non-treated SCR-C, and could not apply as intravitreal injection formulation and control in this study. From these background, we performed the experiments using the LAN-NPs. In order to respond to the reviewer’s comment, we added these contents in the discussion (line 254-259).

Q2. The authors should investigate the effect of concentration of LAN-NPs on the therapeutic efficacy of cataract. Otherwise, the audiences may be curious about why they intended to select specific dose of LAN-NPs for formulation design. The intravitreal injection is a commonly used technique to deliver medications to the posterior segment of the eye.

A2. The reviewer’s comment is correct. In the preliminary study, we evaluated the LAN delivery into the lens by using 3 type administration; eye drops, intracameral injection and intravitreal injection, and the LAN contents in the lens was the highest by the intravitreal injection of LAN-NPs (lanosterol content at 12 h after administration; instillation 1.79 ± 0.15 nmol/lens, intracameral injection 1.99 ± 0.20 nmol/lens, n=3). In addition, the Zhao et al. was also used the intravitreal injection in the in vivo study using dogs (Nature 523: 607-611, 2015, Ref. 3). On the other hand, Zhao et al. was injected the 2 mg/ml lanosterol every 3 days in the dogs, however, the size of lens and vitreous body was different between dog and rat. Taken together, we increased the concentration and number of dose, and determined the dose to 5 mg/ml lanosterol every 2 days in the rats to evaluate whether the high administration of lanosterol repaired the serious structural collapse and opacification in the lenses of SCR-C. In order to respond to the reviewer’s comment, we added the information (line 129-133, 337-339).

Q3. In my opinion, the in vivo fate of injectables is particularly important for clinical end user. The authors simply examine the therapeutic efficacy of the injected LAN-NPs, but do not check the biodistribution of the drugs. It is also necessary to investigate the relationship between in vivo amount of LAN-NPs and residence time.

A3. Thank you for pointing out this. In order to respond to the reviewer’s comment, we measured the plasma LAN levels after the intravitreal injection, and the changes in LAN concentration was not observed. Therefore, the LAN may be distributed in the eye. we added the contents in the result (line 129-130).

Q4. If the authors consider the nanoparticle formulation can improve ocular bioavailability, they should clarify the underlying mechanism. It is also highly desirable to investigate whether the physicochemical properties of nanoparticles play any potential roles in the control of their interactions with biological entities in vivo? The authors should check these parts by comprehensively performing cell-material interaction studies using a series of LAN-NPs with varying sizes and charges.

A4. Thank you very much for pointing this out. It is difficult to prepare the LAN solution without the serious side effect, such as inflammation and lens opacification, since we attempted to prepare the solution containing 0.5% LAN by using the DMSO, ethanol and/or surface active agent, such as tween 80. However, the intravitreal injection of 0.5% LAN solution in DMSO, ethanol and/or surface active agent promote the onset of lens opacification in the SCR-C, and could not apply as intravitreal injection formulation and control. Therefore, we prepared the LAN-NPs. In addition, we evaluated the LAN delivery into the lens by using 3 type administration; eye drops, intracameral injection and intravitreal injection, and the LAN contents in the lens was the highest by the intravitreal injection of LAN-NPs (lanosterol content at 12 h after administration; instillation 1.79 ± 0.15 nmol/lens, intracameral injection 1.99 ± 0.20 nmol/lens, n=3). From these background, we performed the experiments using the LAN-NPs. In order to respond to the reviewer’s comment, we added these contents, and removed the sentence shown about the important of ocular bioabailability, and revised to “In the future, solid nanoparticles may provide a novel strategy for developing safe and effective ophthalmic formulations” in the introduction (line 83-85).

Q5. As stated by the authors, there have been many studies concerning the mechanism of lens opacification in SCR, with results that can be summarized as follows: excessive NO, a product of oxidative stress, causes enhanced lipid peroxidation that results in the oxidative inhibition of Ca2+-ATPase. Please add appropriate citations to support this important claim.

A5. The reviewer’s comment is correct. In order to respond to the reviewer’s comment, we cited the reference 22, 30 and 31 (Reference 22, 30 and 31).

Q6. In the results section, the authors mentioned the zeta potential value and particle size of nanoparticles, but in the discussion section, they did not describe the influence of these parameters on disease treatment.

A6. The reviewer’s comments are very important. The suspensions containing LAN microparticles caused the transient opacity after the intravitreal injection, however it was improved by the decrease in the particle size to nano-order. In fact, the transient opacity was not observed after the intravitreal injection of LAN-NPs in this study. On the other hand, the dispersion stability is important to develop the intravitreal injection formulation, and the zeta potential value is major factor related the dispersion stability. In this study, the zeta value is not enough levels. In fact, the aggregation of LAN particles was not detected for 2 weeks in the 0.5% LAN-NPs, but the LAN particles was aggregated 1 month after the preparation in the 0.5% LAN-NPs. Further studies are needed to enhance the dispersion stability for the useful application. In order to respond to the reviewer’s comment, we added these contents and necessity for enhance the zeta potential value in the discussion (line 240-245, 259-262).

Q7. As stated by the authors, the supply of lanosterol by the intravitreal injection of LAN-NPs repaired the space and structural collapse in the early stages in the lenses of SCR-N and delayed the progress of lens opacification in SCR-C. In order to publish in IJMS, it is highly desirable to investigate the above-mentioned molecular mechanism by designing corresponding experiments.

A7. In this study, the intravitreal injection of LAN-NPs enhanced the LAN contents in the lenses (Fig. 2D), and previous reports showed that LAN disrupts the aggregation of human γD-crystalline by binding to the hydrophobic dimerization interface (Ref. 6). Therefore, these effect may lead to repair the space and structural collapse in the early stages in the lenses of SCR-N. On the other hand, the our previous studies found that the excessive NO via iNOS was increased in the lenses of SCR-C, and the excessive NO decreased Ca2+-ATPase activity by the lipid peroxidation, resulting in enhancement of Ca2+ contents. Furthermore, calpain activity increased by the enhanced Ca2+ contents, and caused the lens opacification. In this study, we measured the calpain activity in the lenses of SCR-C, and the enhancement of calpain activity was observed in the 11-weeks old SCR-C applied with LAN-NPs in comparison with 6-week old SCR-C. These results suggested that the calpain activity via high Ca2+ levels also caused the lens opacification in the SCR-C applied with LAN-NPs. Otherwise, the LAN-NPs attenuated the changes in the cararact-related factors (NO, Ca2+-ATPase, CA2+ and calpain). Taken together, we hypothesized that the space and structure collapse promotes NO production, and that accelerated production of oxidative stress may enhance the lens opacification via calpain activation. In addition, the LAN-NPs attenuated the space and structure collapse in the lenses of SCR-C with aging, resulting in delay the onset of lens opacification via calpain. In order to respond to the reviewer’s comment, we added the data of calpain, and mentioned these contents (line 206-207, 289-291, 313, 320-322, Figure 6E, Reference 22,30,31).

Thank you for great comments.

Round 2

Reviewer 2 Report

The manuscript has improved with the addition of new data and incorporation of suggested revisions. There are a couple minor issues that needs to be addressed.

1) Lines 295 and 296 - This study does not fully support previous reports that  lanosterol prevents cataract formation or clears the lens opacity. The sentence should be revised to indicate this as indicated in revised abstract and discussion. It seems like the authors missed to revise the lines 295-296.

2) crystalline at several places in the MS should be corrected as "crystallin"

Author Response

We carefully revised our manuscript according to the suggestions of the reviewer 2, and details are as follows.

< Q and A for Reviewer 2>

Q1. Lines 295 and 296 - This study does not fully support previous reports that lanosterol prevents cataract formation or clears the lens opacity. The sentence should be revised to indicate this as indicated in revised abstract and discussion. It seems like the authors missed to revise the lines 295-296.

A1. The reviewer’s comments are very important. In order to respond to the reviewer’s comment, we removed the sentence. Thank you very much for pointing this out.

Q2. crystalline at several places in the MS should be corrected as "crystallin".

A2. The reviewer’s comment is correct. We corrected to “crystallin”.

Thank you for great comments.

Reviewer 3 Report

The revised version has adequately addressed most of the critiques raised by this reviewer and maybe suitable for publication in "IJMS".

Author Response

We carefully revised our manuscript according to the suggestions of the reviewer 3, and details are as follows.

<Q and A for Reviewer 3>

Q1. The revised version has adequately addressed most of the critiques raised by this reviewer and maybe suitable for publication in "IJMS".

A1. Thank you very much for reviewing. I am grateful for your detailed and speedy response.

Thank you for great comments.